# In vivo super-resolution RESOLFT microscopy of *Drosophila melanogaster*

**Sebastian Schnorrenberg[1], Tim Grotjohann[1], Gerd Vorbrüggen[2,3], Alf Herzig[2†], Stefan W Hell[1], Stefan Jakobs[1,4]***

[1]Department of NanoBiophotonics, Max Planck Institute for Biophysical Chemistry, Göttingen, Germany; [2]Department of Molecular Developmental Biology, Max Planck Institute for Biophysical Chemistry, Göttingen, Germany; [3]Abteilung Entwicklungsbiologie, Georg-August-Universität Göttingen, Göttingen, Germany; [4]Department of Neurology, University Medical Center of Göttingen, Göttingen, Germany

**Abstract** Despite remarkable developments in diffraction unlimited super-resolution microscopy, *in vivo* nanoscopy of tissues and model organisms is still not satisfactorily established and rarely realized. RESOLFT nanoscopy is particularly suited for live cell imaging because it requires relatively low light levels to overcome the diffraction barrier. Previously, we introduced the reversibly switchable fluorescent protein rsEGFP2, which facilitated fast RESOLFT nanoscopy (*Grotjohann et al., 2012*). In that study, as in most other nanoscopy studies, only cultivated single cells were analyzed. Here, we report on the use of rsEGFP2 for live-cell RESOLFT nanoscopy of sub-cellular structures of intact *Drosophila melanogaster* larvae and of resected tissues. We generated flies expressing fusion proteins of alpha-tubulin and rsEGFP2 highlighting the microtubule cytoskeleton in all cells. By focusing through the intact larval cuticle, we achieved lateral resolution of <60 nm. RESOLFT nanoscopy enabled time-lapse recordings comprising 40 images and facilitated recordings 40 μm deep within fly tissues.

*For correspondence: sjakobs@gwdg.de

**Present address:** [†]Department of Cellular Microbiology, Max Planck Institute for Infection Biology, Berlin, Germany

## Introduction

The various super-resolution fluorescence microscopy (nanoscopy) methods that allow overcoming the diffraction barrier have become an indispensable part of the modern biology toolbox (*Huang et al., 2009*). Still, entire research fields such as those focusing on the model organism *Drosophila melanogaster* have been reluctant in adapting live-cell nanoscopy. Indeed, nanoscopy of thick specimen is mostly performed on optically transparent samples such as cleared and fixed tissue (*Stelzer, 2015*). The relatively few reported examples of nanoscopy in living tissues typically relied on specimen in which individual cells overexpressed a fluorescent protein (*Nägerl et al., 2008*; *Chéreau et al., 2015*; *Berning et al., 2012*; *Yuste and Bonhoeffer, 2004*; *Testa et al., 2012*; *Tønnesen et al., 2014*).

To overcome this shortcoming, we previously reported the generation of rsEGFP2 (*Grotjohann et al., 2012*), a very photostable reversibly switchable fluorescent protein (RSFP). The green fluorescent rsEGFP2 can be reversible photo-switched between a fluorescent 'On-' and a non-fluorescent 'Off-'state with light of ~405 nm and ~488 nm. rsEGFP2 was designed to be used within the framework of live-cell RESOLFT (reversible saturable optical fluorescence transition) nanoscopy (*Hell et al., 2004*; *Brakemann et al., 2011*; *Grotjohann et al., 2011*). In its initial application, however, rsEGFP2 was only used to study sub-cellular dynamics in single-layered cultivated cells (*Grotjohann et al., 2012*).

As detailed elsewhere (*Hell, 2009*), in RESOLFT nanoscopy, a light pattern such as a 'doughnut' featuring an intensity zero at is center is scanned across the sample. Several variations of RESOLFT microscopy have been established, including single-beam and parallelized scanning approaches (*Brakemann et al., 2011*; *Chmyrov et al., 2013*; *Grotjohann et al., 2011*). The RESOLFT principle using RSFPs has also been extended to nonlinear structured illumination microscopy (PA NL-SIM) (*Li et al., 2015*) and light sheet microscopy (*Hoyer et al., 2016*). In RESOLFT, the scanned light pattern typically induces the transition into the Off-state, so that the On-state molecules are confined to a sub-diffraction volume. RESOLFT nanoscopy stands out from all other far-field super-resolution microscopies that overcome the diffraction barrier by the relatively low light dose that is required to achieve nanoscale resolution. The light intensities used are up to six orders of magnitude lower than those in STED-microscopy (*Klar et al., 2000*) and are comparable to those typically applied in live-cell confocal fluorescence microscopy (*Wäldchen et al., 2015*). Likewise, the total light dose impinging on the sample is lower by 3–4 orders of magnitude compared to stochastic single-molecule based nanoscopy approaches (*Grotjohann et al., 2012*). Furthermore, when implemented in a beam scanning approach, RESOLFT can be combined with confocal detection, which is particularly suitable for the imaging of highly fluorescent tissues because it rejects the out-of-focus fluorescence. Altogether, this suggests that RESOLFT microscopy is particularly suitable for live-cell imaging of complex samples such as tissues.

Still, to our knowledge, neither RESOLFT nor any other diffraction-unlimited super-resolution microscopy approach has so far been used to image subcellular details in living transgenic animals ubiquitously expressing a fusion protein. In this study we demonstrate live-cell RESOLFT nanoscopy on living transgenic fly tissues and intact larvae whose microtubule network was labelled in all cells by ubiquitously expressing rsEGFP2 fused to α-tubulin.

## Results

### Generation of transgenic flies expressing rsEGFP2-α-tubulin

In order to enable RESOLFT nanoscopy of living *Drosophila melanogaster*, we generated flies expressing rsEGFP2-α-tubulin ubiquitously under the transcriptional control of the ubiquitin 68E promotor ensuring moderate expression levels of the fusion protein. Confocal images of resected tissues from transgenic adult flies revealed a large variation of the arrangement of the microtubule cytoskeleton depending on the cell type (*Figure 1A–J*). At 25°C, the half-time of the life span of these flies was ~41 days, similar to the isogenic wild type strain. The flies were propagated for more than 50 generations without detecting any loss in fitness. The fact that even during oogenesis and spermatogenesis the microtubule network was labelled (*Figure 1D,E*) suggests no adverse effects of the ubiquitous expression of rsEGFP2-α-tubulin on the embryogenesis and demonstrates the versatility of rsEGFP2 as a live-cell label. Western blots of larval protein extracts probed with antiserum against α-tubulin demonstrated that the fusion protein was expressed at moderate levels, so that the ratio of endogenous α-tubulin to rsEGFP2-α-tubulin was about 3:1 (*Figure 1—figure supplement 1*).

Together, we generated a stable transgenic fly line ubiquitously expressing rsEGFP2-α-tubulin in all tissues specifically highlighting the microtubule cytoskeleton without interfering with the fly fitness.

### RESOLFT nanoscopy of resected fly tissues

In order to investigate the use of RESOLFT nanoscopy for imaging subcellular structures in living resected fly tissues, we prepared various tissues from adult files and maintained the tissues in Schneider's cell culture medium under the coverslip to ensure the viability of the cells. Images were recorded using a beam-scanning RESOLFT microscope. To unambiguously report the potential and current limitations of beam scanning RESOLFT, all microscopic images display only raw data with linear color maps and no background subtraction throughout the manuscript. All color maps including the actual photon counts are given in *Supplementary file 1*. Deconvolution or any other mathematical image processing was not applied. To fairly compare a RESOLFT image with the best possible corresponding diffraction-limited image, we recorded the confocal image together with the RESOLFT image using the same pixel sizes and recording times by just omitting the switching off-step. We

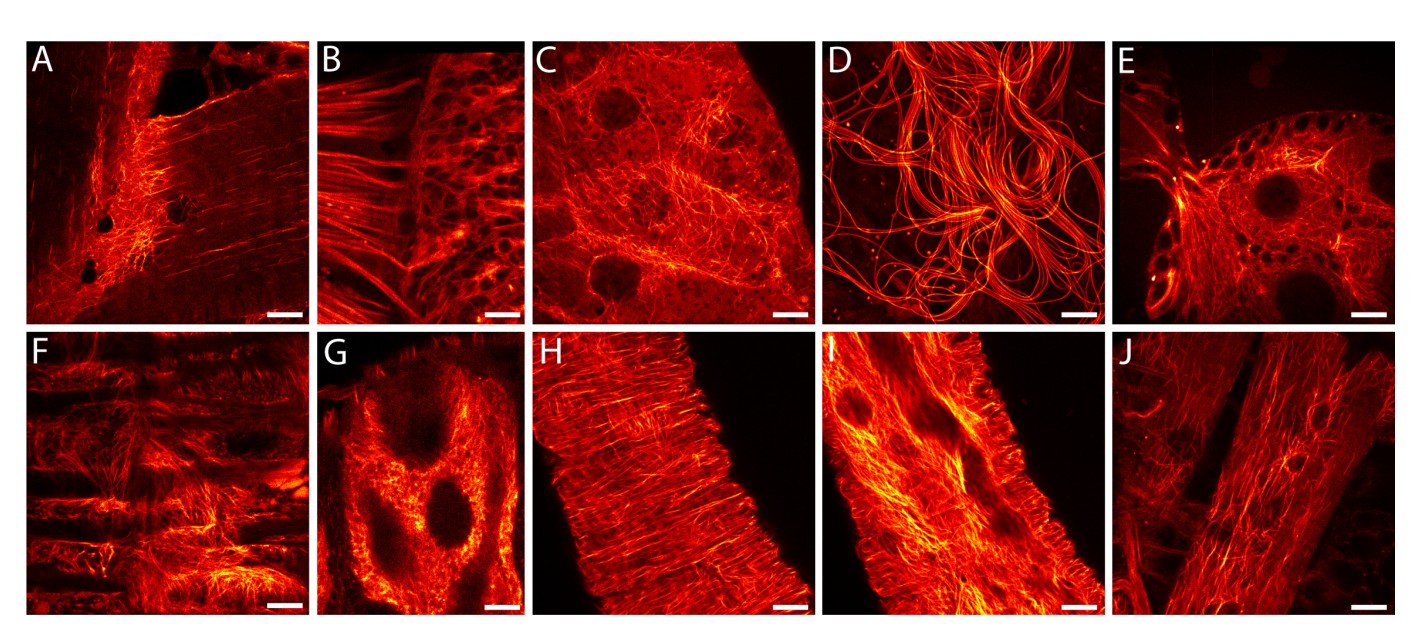

**Figure 1.** *Drosophila melanogaster* ubiquitously expressing rsEGFP2-α-tubulin. Confocal recordings of living resected tissues of transgenic wandering third instar larvae (**A–C**, **F–J**) and of adult flies (**D,E**) ubiquitously expressing rsEGFP2-α-tubulin. (**A**) Body wall attachment site, (**B**) eye imaginal disc, (**C**) salivary glands, (**D**) sperm, (**E**) ovaries, (**F**) intestinal muscles, (**G**) intestine, (**H**) salivary duct, (**I**) salivary duct, (**J**) body wall muscles. Images display raw data. Scale bars: 10 μm.

The following figure supplement is available for figure 1:

**Figure supplement 1.** Western Blot analysis of the expression level of rsEGFP2-α-tubulin in *Drosophila* larvae.

recorded RESOLFT images of various tissues including body wall muscles (*Figure 2A–C*) and salivary glands (*Figure 2—figure supplement 1*), demonstrating a distinct improvement in resolution compared to the corresponding confocal images. The increased resolution is well exemplified in body wall muscle cells, where microtubules are frequently co-aligned over relatively long distances (*Figure 2B, C*).

In order to quantify the attained resolution in the RESOLFT images, we measured the diameter of the rsEGFP2-α-tubulin labeled microtubules by averaging three (*Figure 2—figure supplement 1*) or ten (*Figure 3—figure supplement 1*) neighboring intensity profiles, each separated by the edge length of a pixel, across a single filament. This approach excludes the occurrence of erroneous small resolution values, which otherwise may arise from statistical outliers in the signal intensity of individual pixels. The full width at half maximum (FWHM) of these line profiles was determined by fitting a Lorentz curve to the measured data. The FWHM values were consistently around 50–60 nm (*Figure 2—figure supplement 1B*; *Figure 3—figure supplement 1*), which represents an ~4-fold improvement over the diffraction limited resolution.

So far, the RESOLFT microscope was used in the 2D mode, i.e. the optical resolution was improved only in the lateral plane, whereas in the axial direction it was diffraction limited. The confocal detection allows optical sectioning of the tissue. To make use of this property, we recorded 14 RESOLFT images of resected body wall muscle tissue with 250 nm axial distance between each image. Thereby one can track individual microtubules in the tissue volume (*Figure 2—figure supplement 2*; *Video 1*), albeit at relatively poor axial resolution. In order to track the microtubules more reliably in 3D, we replaced the 2D-doughnut by a 3D-doughnut, which confines the On-state rsEGFP2 molcules in all room directions, thus increasing the resolution along the x,y, and z-axes. Using this we imaged 33 RESOLFT images in a 2 μm thick volume element with an ~4 fold improved axial resolution (*Figure 2—figure supplement 3*; *Video 2*).

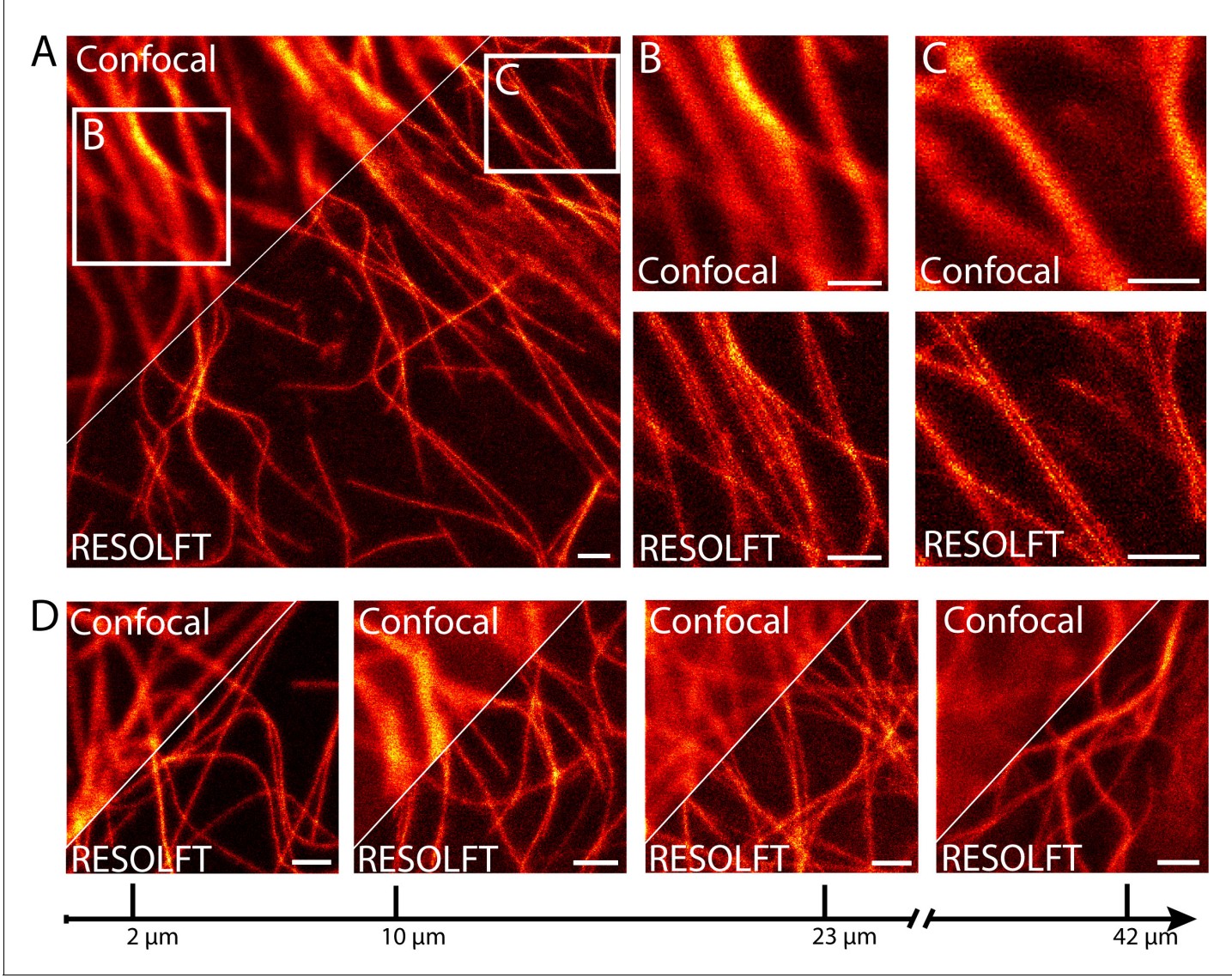

**Figure 2.** RESOLFT nanoscopy of resected tissues of third instar larvae. (**A**) Comparison of confocal and RESOLFT recordings taken on body wall muscles of a larva expressing rsEGFP2-α-tubulin. (**B,C**) Magnifications of the areas indicated in (**A**). (**D**) RESOLFT and corresponding confocal images were taken at the indicated depths on body wall muscles of a dissected third instar larva ubiquitously expressing rsEGFP2-α-tubulin. All images display raw data. Scale bars: 1 μm.

The following figure supplements are available for figure 2:

**Figure supplement 1.** RESOLFT nanoscopy of resected tissues of third instar larvae.

**Figure supplement 2.** RESOLFT volume imaging using a 2D-doughnut for off-switching.

**Figure supplement 3.** RESOLFT volume imaging using a 3D-doughnut for off-switching.

We conclude that RESOLFT nanoscopy facilitates sub-diffraction imaging in living resected tissues to visualize structures not accessible by conventional microscopy.

## RESOLFT imaging within the tissue

Imaging resected tissues has the advantage that the cells of interest are typically in the top layers of the tissue, simplifying the recordings. In order to evaluate the strength of 2D-RESOLFT microscopy

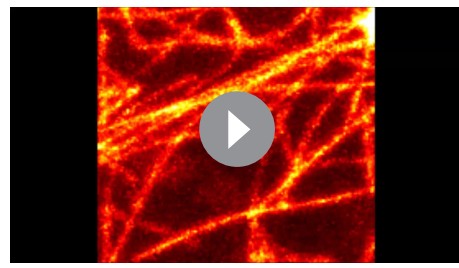

**Video 1.** Animation of a 3D stack recorded with a RESOLFT microscope featuring a 2D-doughnut for off-switching. The axial resolution was diffraction limited (confocal detection). Dissected living body wall muscles of a *Drosophila melanogaster* larva ubiquitously expressing rsEGFP2-α-tubulin were imaged. The imaged volume has a size of 4.2 μm x 3.9 μm x 3.5 μm. We recorded 14 RESOLFT images with 250 nm axial distance between each image. The 3D reconstruction was generated using the program ImageJ with the plugin 3D Project. See *Figure 2— figure supplement 2*.

for imaging of sub-cellular structures buried in a highly fluorescent and scattering tissue, a wandering third instar larvae ubiquitously expressing rsEGFP2-α-tubulin was cut into half and the inverted front half of the larva was mounted in physiological Schneider's cell culture medium. We equipped the microscope with a silicone oil immersion objective in order to match the refractive index of the tissue. Still, the confocal images exhibited reduced signal-to-noise ratios already at an imaging depth of 10 μm due to the pronounced scattering in the tissue. This undesirable effect strengthened with increasing imaging depths such that at an imaging depth of 42 μm the rsEGFP2-α-tubulin labeled microtubules were barely discernible when imaged in the confocal mode (*Figure 2D*).

Using RESOLFT nanoscopy, the highest attainable resolution and the best signal-to-noise ratio were recorded in the top layer of the opened larvae (*Figure 2D*). However, also at deeper cell layers, we resolved individual microtubule filaments with a substantially better signal-to-noise ratio than in the confocal case. We attribute this advantage of RESOLFT to the beneficial effect of transferring the rsEGFP2 molecules outside of the focus to a non-fluorescent state before probing the fluorescence, thus improving the contrast. The resolution decreased with the penetration depth, but was always better than in the corresponding confocal images. Hence the RESOLFT mode allows retrieving image information deep within scattering and highly fluorescent tissue that is inaccessible by confocal microscopy.

## In vivo RESOLFT nanoscopy of larvae

Living intact *Drosophila* larvae are difficult samples for fluorescence microscopy, not only because of their challenging optical properties, but also because of their frequent irregular movements and the cuticle induced light scattering. Moreover, in samples that express the fusion proteins in all cells,

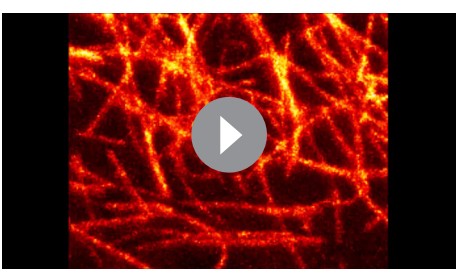

**Video 2.** Animation of a 3D stack recorded with a RESOLFT microscope featuring a 3D-doughnut for off-switching. This configuration of the microscope improved the optical resolution along all three directions. Dissected living body wall muscles of a *Drosophila melanogaster* larva ubiquitously expressing rsEGFP2-α-tubulin were imaged. The imaged volume has a size of 8.5 μm x 6.8 μm x 2 μm. We recorded 33 RESOLFT images with 60 nm axial distance between each image. The 3D reconstruction was generated using the program ImageJ with the plugin 3D Project. See *Figure 2—figure supplement 3*.

very pronounced out-of-focus fluorescence may occur. To enable RESOLFT microscopy in intact larvae, living rsEGFP2-α-tubulin expressing second instar larvae were placed between two spacers that separated the object slide and the cover glass, forming a cavity filled with Schneider's cell culture media (*Figure 3A*). By this, the overall movements of the larvae were largely restrained, although movements of body muscles still occurred. For RESOLFT imaging, we focused through the intact larval cuticle into the underlying tissue without any further treatment of the larvae. Despite the occasional movements of the larvae, we could reliably record RESOLFT images of the microtubule network in these intact animals. Co-aligned microtubules that were fully blurred in the corresponding confocal images could be resolved (*Figure 3B–D*). As in resected tissues, the measured resolution in the intact larvae was also ≤60 nm (*Figure 3E*; *Figure 3—figure supplement 1*), despite the fact that we were focusing through the opaque

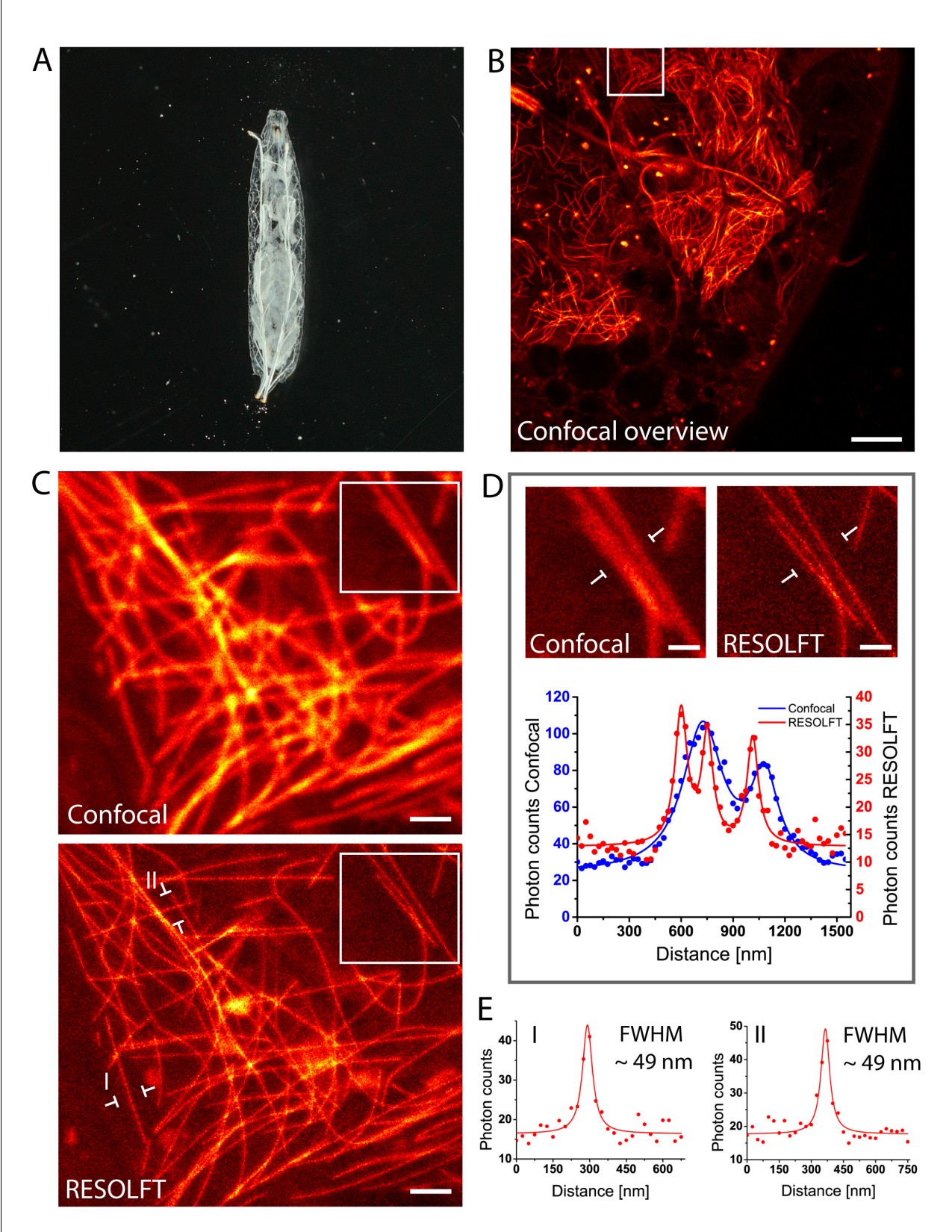

**Figure 3.** In vivo RESOLFT imaging of intact living *Drosophila melanogaster* larvae. (A) Living second instar larva expressing rsEGFP2-α-tubulin were placed under a coverslip between two spacers in Schneider's medium. (B) Confocal overview imaged through the cuticle of the larva. (C) Confocal (top)
*Figure 3 continued on next page*

*Figure 3 continued*

and corresponding RESOLFT (bottom) image of the area indicated in (B). (D) Top: Magnifications of the areas indicated in (C). Bottom: Line profiles taken at the indicated positions. The data points represent an average of three adjacent (25 nm distance) measurements. (E) Line profiles taken at the positions indicated in (C). The data points represent an average of three adjacent (25 nm distance) measurements. The averaged data were fitted with a Lorentzian function (solid line). The FWHM was determined on the fitted function. See also (*Figure 3—figure supplement 1*). Images display raw data. Scale bars: 10 μm (B), 1 μm (C), and 500 nm (D).

The following figure supplement is available for figure 3:

**Figure supplement 1.** Determination of the achieved resolution in intact larvae.

cuticle into tissue. We conclude that RESOLFT imaging is feasible even on living larvae expressing rsEGFP2-fusion proteins without removing the cuticle and thus leaving the larvae entirely intact.

## Time-lapse RESOLFT nanoscopy

Next, we imaged the dynamical change of the microtubule network over time with high spatial resolution. Over a large field of view of 552 μm$^2$, we imaged dissected larval body wall muscles continuously for 2.5 hr in the RESOLFT mode, thereby recording 40 RESOLFT images (*Figure 4A*, *Video 3*). At the end of the recording, the microtubules still displayed alternating periods of growth and shrinkage. During this extended imaging period we observed no obvious signs of cellular distress, neither in the recorded area nor in the surrounding cells, despite recognizable photobleaching (*Figure 4—figure supplement 1*). To follow the growth and depolymerization of individual microtubules, we reduced the field of view to 52 μm$^2$ and adapted the imaging conditions to record a single RESOLFT image in 9.3 s (*Figure 4B*, *Video 4*). At this frame rate we recorded 40 RESOLFT images, facilitating the tracking of microtubule dynamics. Hence RESOLFT microscopy can be utilized for time-lapse recordings of dynamic sub-cellular processes in highly fluorescent tissues.

## Discussion

In previous RESOLFT studies, live cell recordings of only single cells were reported (*Tiwari et al., 2015*, *Duwé et al., 2015*; *El Khatib et al., 2016*; *Brakemann et al., 2011*; *Grotjohann et al., 2011*). We extended the use of rsEGFP2 (*Grotjohann et al., 2012*) to in vivo RESOLFT imaging of tissues and larvae.

During this study we did not observe detrimental effects of the ubiquitous expression of rsEGFP2-α-tubulin fusion proteins to the flies. Importantly, we also did not observe phototoxic effects during imaging. We attribute this to the relatively low light intensities required in RESOLFT nanoscopy to overcome the diffraction barrier. Moreover, because of the utilized scanning approach, the individual irradiation times are short and consequently the overall light dose deposited in the imaged volume is small. Nonetheless, the light intensities in RESOLFT microscopy are still higher than what is sometimes regarded as critical intensity threshold in long-term imaging to follow developmental processes over hours or even days (*Stelzer, 2015*). It should be noted, however, that in in vivo RESOLFT microscopy, typically only a very small part of the sample organism is imaged, whereas the remaining parts are not illuminated. Generally, the phototoxic effects that manifest themselves after the actual experiment are irrelevant as long as the cellular dynamics during the imaging are not disturbed.

The relatively low light intensities required for RESOLFT nanoscopy and the flexibility of the method offer additional freedom in the choice between resolution and recording time: In RESOLFT nanoscopy, the attained resolution can be adjusted so that, for example, the resolution is decreased in favor of reduced phototoxicity, increased recording speed, or the number of images taken before bleaching. Although in this manuscript we only show raw image data, the RESOLFT data could be deconvolved to increase the contrast and resolution even further if required. As shown here, the RESOLFT modality offers the benefit of allowing to image deeper in tissue. This observation relys on the fact that fluorescence from outside the focal region is precluded because the molecules located in that region are transferred into a non-fluorescent state; their signal cannot perturb the image.

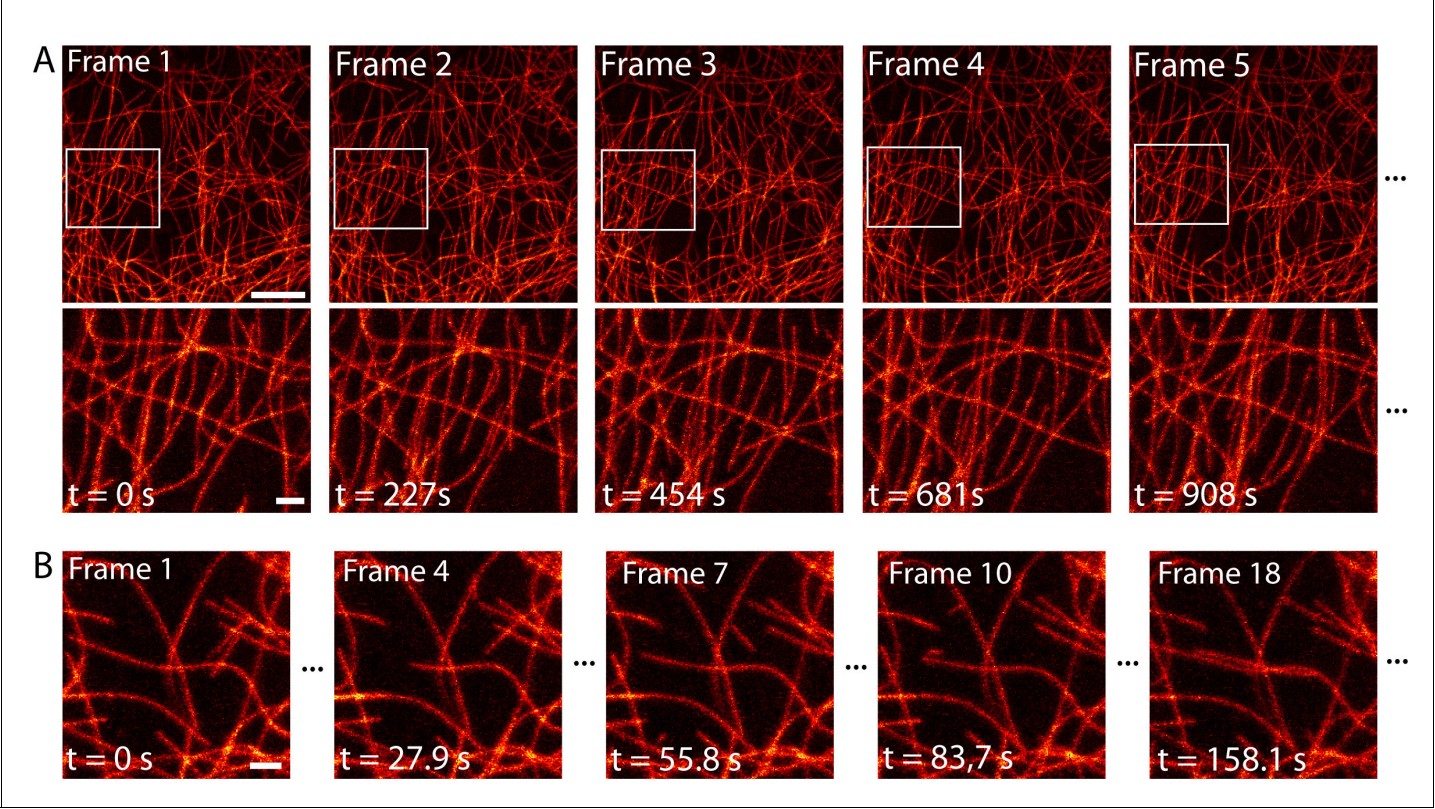

**Figure 4.** Time-lapse RESOLFT imaging of the microtubule cytoskeleton in body wall muscles of dissected *Drosophila melanogaster* larvae. (**A**) First five of in total 40 RESOLFT image frames of dissected larva tissue expressing rsEGFP2-α-tubulin. Images were continuously recorded. Recording time of one frame: 227 s. Top: RESOLFT overview; bottom: Magnifications of the areas indicated. (**B**) Time lapse RESOLFT imaging with high frame rate. Selected frames of in total 40 frames recorded on body wall muscles are displayed. Recording time of one frame: 9.3 s. Data are also represented in *Videos 3* and *4*. All images display raw data. Scale bars: 5 μm (**A**, top row) and 1 μm (**A**, bottom row, and **B**).

The following figure supplement is available for figure 4:

**Figure supplement 1.** Confocal overview of the region shown in *Figure 4A* and *Video 3* before and after long-term RESOLFT time-lapse imaging.

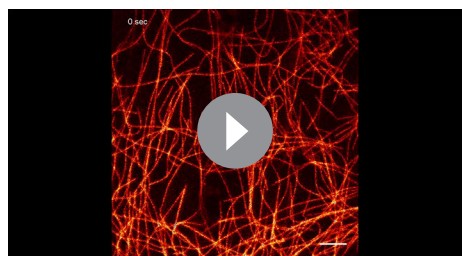

**Video 3.** Time-lapse RESOLFT imaging of the microtubule cytoskeleton in dissected *Drosophila melanogaster* third instar larvae body wall muscles. 40 RESOLFT image frames were continuously recorded. Recording time of one frame: 227 s. The video displays 4 frames per second. Corresponding still images are shown in *Figure 4A*. Scale bar: 2.5 μm.

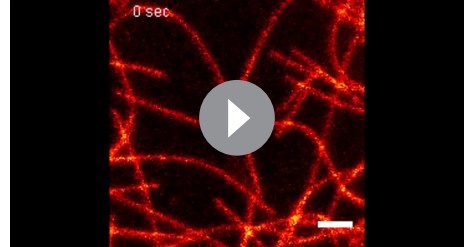

**Video 4.** Time-lapse RESOLFT imaging of the microtubule cytoskeleton in dissected *Drosophila melanogaster* third instar larvae body wall muscles. 40 RESOLFT image frames were continuously recorded. Recording time of one frame: 9.3 s. The video displays 4 frames per second. Corresponding still images are shown in *Figure 4B*. Scale bar: 1 μm.

Since the release of rsEGFP2 in 2012 (*Grotjohann et al., 2012*), a number of novel switchable fluorescent proteins have been reported. Of these, the rsEGFP/rsEGFP2 variants rsFolder (*El Khatib et al., 2016*), rsGreen (*Duwé et al., 2015*), and rsEGFP-N205S (*Chmyrov et al., 2013*), the red fluorescent rsCherryRev1.4 (*Lavoie-Cardinal et al., 2014*), and the positive switching Kohinoor (*Tiwari et al., 2015*) have been successfully used in RESOLFT nanoscopy. Also organic dyes were used in this concept (*Kwon et al., 2015*). Still, for the applications shown in this manuscript, namely beam scanning RESOLFT nanoscopy in tissues, rsEGFP2 seems to unite the necessary properties in the most suitable combination. Although rsEGFP2 dramatically outperforms earlier RSFPs such as Dronpa (*Ando et al., 2004*) or rsEGFP (*Grotjohann et al., 2011*), we expect generation of superior RSFPs in the future that are even more photostable, brighter, and exhibit better contrast.

We conclude that with the ongoing developments both on RSFPs as well as on instrumentation, it can be anticipated that the application range of low light RESOLFT nanoscopy in *Drosophila* as well as in other multicellular organism will be further expanded in the future.

## Materials and methods

### Cloning and fly generation

A synthetic gene sequence encoding the fluorescent protein rsEGFP2 (GenBank #AGE84598.1) attached to the N-terminus of the α-tubulin 84B (cg1913, UniProt #P06603) protein was synthesized. The DNA was cloned into an attB-containing fly transformation vector under the control of the Ubiquitin p63E promotor-region (3L; 3901762 to 3903743). To generate transgenic flies, a standard phiC31 integrase based germ line transformation procedure was followed, targeting the landing site *M{3xP3-RFP.attP}ZH-86Fb* using Bloomington stock #24749 (*Bischof et al., 2007*).

### RESOLFT and conventional microscopy

RESOLFT nanoscopy was performed using a modified 1C RESOLFT QUAD Scanning microscope (Abberior Instruments, Goettingen, Germany). For RESOLFT imaging of *Drosophila melanogaster*, the following switching scheme was applied to the sample at each scanning position: First, rsEGFP2 proteins were switched into the on-state with light of 405 nm. Second, rsEGFP2 proteins in the periphery of the focal spot were switched off using a 488 nm doughnut-shaped beam. The 2D-doughnut was realized by using a phase plate, whereas the 3D-doughnut was generated by using a spatial light modulator. Third, on-state fluorophores at the center of the spot were read out with a Gaussian shaped beam of 488 nm light. We introduced a short 5 µs break between each step. The RESOLFT images were recorded with or without line accumulation depending on the signal intensity. A detailed listing of the laser powers, switching times, line accumulations and scanning step sizes of all RESOLFT images shown are provided in *Supplementary file 2*. The corresponding confocal images were recorded by applying the same switching scheme as used for the RESOLFT imaging without the illumination step using a doughnut shaped beam (step 2).

### Data analysis and image manipulation

All fluorescence images presented in this work are raw data, neither background subtraction nor any other image data processing has been applied. For the determination of the FWHM, three or ten averaged adjacent line profiles were fitted (standard Lorentz-fit) using Origin 9.1 software.

### Sample preparation

Tissues from wandering third instar larvae were dissected in Schneider's cell culture medium (Life Technologies, Carlsbad, California USA) supplemented with 10% Fetal Bovine Serum (FBS) and mounted under a coverslip in Schneider's cell culture medium supplemented with 10% FBS. For imaging of intact larvae, second instar larvae were placed between two spacers in Schneider's medium and covered with a coverslip. The samples were sealed using nontoxic duplicating silicone (Picodent, Wipperfuerth, Germany).

### Western blotting

Wandering third instar larvae were dissected in 1x PBS to remove the fat tissue before they were homogenized in protein extraction buffer (10% Glycerol, 50 mM HEPES (pH 7,5), 150 mM NaCl, 0,5%

Triton-X-100, 1,5 mM MgCl$_2$, 1 mM EGTA) containing protease inhibitors. Lysate was centrifuged to pellet cellular debris. The supernatant was transferred to a new Eppendorf tube and mixed with Laemmli Buffer (Sigma-Aldrich, St. Louis, Missouri, USA) and denatured for 10 min at 95°C before loading onto a SDS-polyacrylamide gel. The protein-lysate of approximately two larvae was loaded into each lane of the gel. The SDS gel was blotted using a semi-dry blotter onto a PVDF membrane. The protein-blot was blocked using 5% fat free milk powder dissolved in 1x PBS for 1 hr. The polyclonal α-tubulin antiserum (Abcam, Cambridge, England) and the anti-GFP antibody (Clonetech) were used in a concentration of 1:5000 in 2,5% fat free milk powder dissolved in 1x PBS for 1 hr. HRP-conjugated secondary antibody (Jackson ImmunoResearch, Suffolk, UK) was used at a dilution of 1:10.000 in 2,5% fat free milk powder dissolved in 1x PBS for 1 hr and detected with an ECL-kit (Perkin Elmer Life Science, Massachusetts, USA) using a custom built imaging system with a CCD-camera.

## Acknowledgements

We thank Herbert Jäckle, MPIbpc, for supporting our fly work. We acknowledge Jaydev Jethwa for carefully reading the manuscript. This work was supported by the Cluster of Excellence and DFG Research Center Nanoscale Microscopy and Molecular Physiology of the Brain (to SJ).

## Additional information

### Competing interests

SWH: The Max Planck Society and SWH hold patent rights (US 7,064,824) on the RESOLFT principle. The other authors declare that no competing interests exist.

### Funding

| Funder | Author |
| --- | --- |
| Deutsche Forschungsgemeinschaft | Stefan Jakobs |
| Max-Planck-Gesellschaft | Gerd Vorbrüggen<br>Alf Herzig<br>Stefan W Hell<br>Stefan Jakobs |

The funders had no role in study design, data collection and interpretation, or the decision to submit the work for publication.

### Author contributions

SS, Conception and design, Acquisition of data, Analysis and interpretation of data, Drafting or revising the article; TG, GV, AH, SWH, Analysis and interpretation of data, Drafting or revising the article; SJ, Conception and design, Analysis and interpretation of data, Drafting or revising the article

### Author ORCIDs

Stefan Jakobs, http://orcid.org/0000-0002-8028-3121

## Additional files

### Supplementary files

• Supplementary file 1. Representation of color maps used. The numbers indicate the photon counts that correspond to the lowest and highest value of the respective color map. Blue color indicates saturation.

• Supplementary file 2. Imaging parameters used for RESOLFT microscopy. Light powers were measured in front of the objective's back focal plane.

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
