## [Decision Letter]

Thank you for submitting your article "In vivo super-resolution RESOLFT microscopy of *Drosophila melanogaster*" for consideration by *eLife*. Your article has been reviewed by two peer reviewers, and the evaluation has been overseen by a Reviewing Editor and Detlef Weigel as the Senior Editor. One of the two reviewers, Kyu Young Han, has agreed to reveal his identity.

The reviewers have discussed the reviews with one another and the Reviewing Editor has drafted this decision to help you prepare a revised submission.

Summary:

Current fluorescence nanoscopy methods allow visualizing sub-cellular structures with unprecedented resolution. However, in vivo fluorescence nanoscopy has been challenging because it requires fast imaging speed and minimal photodamage. In this aspect, RESOLFT nanoscopy is a very promising tool due to its low dosage of illumination although its imaging speed is somewhat slower than STED microscopy. RESOLFT uses lower doses than STED as well as PALM/STORM techniques and filters out out-of-focus light with the pinhole of the confocal scanning geometry. This combination, in concert with fluorescent proteins, makes it the best-suited super-resolution technique for in vivo imaging. In this manuscript, the authors present significant advances over their 2012 *eLife* publication on the development of rsEGFP2 as a genetically encodable RESOLFT probe and its application in live-cell RESOLFT nanoscopy.

They demonstrate (i) a successful preparation of transgenic *Drosophila melanogaster* stably expressing RSFP-tubulin, and (ii) the extension of applications of RESOLFT nanoscopy to fruit fly tissue sections, deep tissue imaging (> 40 um) and imaging of intact embryo, which are almost impossible for most of fluorescence nanoscopy. The manuscript is written concisely, is easy to understand and the results are presented transparently. The experiments were executed beautifully. The work represents an important milestone in the development of super-resolution microscopy.

Essential revisions:

1) We understand 3D imaging is not a main part of this manuscript. However, a volumetric imaging is essential to visualize three-dimensional distributions of fluorescent markers of interest in tissues. We request adding 3D RESOLFT images of small volume as supplementary data. If it is not viable, please specify its reason in the main text.

2) Since the introduction of RSFP by Grotjohann in 2011, there has been significant progress in the development of diverse RSFP. The authors need to elaborate on this at Discussion section.

3) In terms of temporal and spatial resolution, RESOLFT nanoscopy is in competition with the recently demonstrated photoactivatable nonlinear SIM (PA NL-SIM) microscopy [Li et al., Science (2015)]. Although they imaged cultured cells in that paper, it looks straightforward to apply it for tissue imaging. How would you compare the current RESOLFT nanoscopy with PA NL-SIM microscopy for in vivo imaging? It would be great to briefly address it in the main text.

---

## [Author Response]

1) We understand 3D imaging is not a main part of this manuscript. However, a volumetric imaging is essential to visualize three-dimensional distributions of fluorescent markers of interest in tissues. We request adding 3D RESOLFT images of small volume as supplementary data. If it is not viable, please specify its reason in the main text.

We are thankful for this suggestion. We added two new data sets to the revised version of the manuscript to demonstrate 3D RESOLFT:

A) In all images shown in the original manuscript we used a 2D doughnut that facilitated improved lateral resolution and conventional confocal axial resolution. Of course, the latter allows optical sectioning. In the revised version of the manuscript we demonstrate that this property can be used to follow individual microtubules in space. To this end, we imaged a tissue volume of 4.2 µm x 3.9 µm x 3.5 µm and took 14 xy RESOLFT images, each separated by 250 nm. This allowed a 3D rendering of the tissue section (Figure 2—figure supplement 2; Video 1).

B) Following this suggestion by the reviewers further, we re-designed our microscope and implemented a spatial light modulator (SLM). The SLM allows to generate a 3D doughnut, enabling a resolution of ~140 nm along the z-axis. Using this 3D doughnut we imaged a volume of 8.5 µm x 6.8 µm x 2 µm dimensions. We recorded 33 xy-images, each separated by 60 nm. This data set shows that it is indeed possible to record 3D RESOLFT images on living tissues that have an improved resolution in all directions. These new data are represented in the new Figure 2—figure supplement 3 and in Video 2.

2) Since the introduction of RSFP by Grotjohann in 2011, there has been significant progress in the development of diverse RSFP. The authors need to elaborate on this at Discussion section.

Again, we are thankful for this suggestion. In the revised version of the manuscript we have added a new paragraph to the Discussion to highlight new developments in the field of RSFPs for RESOLFT.

It reads: “Since the release of rsEGFP2 in 2012 (Grotjohann et al. 2012), a number of novel switchable fluorescent proteins have been reported. Of these, the rsEGFP/rsEGFP2 variants rsFolder (El Khatib et al. 2016), rsGreen (Duwe et al. 2015), and rsEGFP-N205S (Chmyrov et al. 2013), the red fluorescent rsCherryRev1.4 (Lavoie-Cardinal et al. 2014), and the positive switching Kohinoor (Tiwari et al. 2015) have been successfully used in RESOLFT nanoscopy. Also organic dyes were used in this concept (Kwon et al. 2015). Still, for the applications shown in this manuscript, namely beam scanning RESOLFT nanoscopy in tissues, rsEGFP2 seems to unite the necessary properties in the most suitable combination. Although rsEGFP2 dramatically outperforms earlier RSFPs such as Dronpa (Ando, Mizuno, and Miyawaki 2004) or rsEGFP (Grotjohann et al. 2011), we expect generation of superior RSFPs in the future that are even more photostable, brighter, and exhibit better contrast.”

*3) In terms of temporal and spatial resolution, RESOLFT nanoscopy is in competition with the recently demonstrated photoactivatable nonlinear SIM (PA NL-SIM) microscopy [Li et al., Science (2015)]. Although they imaged cultured cells in that paper, it looks straightforward to apply it for tissue imaging. How would you compare the current RESOLFT nanoscopy with PA NL-SIM microscopy for in vivo imaging? It would be great to briefly address it in the main text.*

The publication by Eric Betzig and colleagues is an important and also controversially discussed manuscript. Indeed, the shown RESOLFT microscopy and PA NL-SIM are highly related as both approaches rely on the reversible switching of RSFPs to overcome the diffraction barrier. However, the technical implementations are different, as PA NL-SIM relies on structured illumination and widefield detection and our RESOLFT approach on focused excitation and confocal detection. Therefore, we think that it would be rather challenging to implement PA NL-SIM for highly fluorescent tissues.

Two of the authors of this paper (TG and SJ) have contributed to a Technical Comment on the manuscript by Li et al. that was recently published in Science (Science, 2016, 352 (6285): 527); Li and Betzig responded in Science (Science, 2016, 352 (6285): 527). We think that it would not be appropriate to extend this discussion into this paper. Therefore, we prefer to cite the manuscript by Li et al., and to refer readers to the original manuscript by Li et al. without contributing to this ongoing scientific discussion within this publication.

In the revised version of our manuscript it reads: “Several variations of RESOLFT microscopy have been established, including single-beam and parallelized scanning approaches (Brakemann et al. 2011, Chmyrov et al. 2013, Grotjohann et al. 2011). The RESOLFT principle using RSFPs has also been extended to nonlinear structured illumination microscopy (PA NL-SIM) (Li et al. 2015) and light sheet microscopy (Hoyer et al. 2016).”